# Differences by sexual orientation in treatment outcome and satisfaction with treatment among inpatients of a German psychiatric clinic

**Martin Plöderl**[1]*, **Robert Mestel**[2], **Clemens Fartacek**[1]

**1** Department for Crisis Intervention and Suicide Prevention and Department for Clinical Psychology, University Clinic for Psychiatry, Psychotherapy, and Psychosomatics, Paracelsus Medical University, Salzburg, Austria, **2** VAMED Clinic Bad Grönenbach, Bad Grönenbach, Germany

* m.ploederl@salk.at

## Abstract

A wealth of research suggests that sexual minority individuals experience stigma and lack of sexual minority specific competencies in mental health care, which could lead to less optimal treatment outcome. However, most related research suffers from methodological limitations, such as selected samples, retrospective design, or not assessing treatment outcome. To overcome some of these limitations, we explored if sexual minority patients have poorer treatment outcome and are less satisfied with treatment in a mental health care setting not specialized in sexual minority issues. The analytical sample comprised 5609 inpatients, including 11% sexual minority patients, from a German psychiatric clinic. Outcomes were improvement in well-being and depression from admission to discharge, and satisfaction with treatment judged at discharge. Nearly all sexual orientation differences were in a direction hinting at less improvement of depression and well-being and less satisfaction among sexual minority compared to heterosexual patients. However, the differences were generally small and not statistically significant. Stigma and lacking sexual orientation specific competency in healthcare may not be universally present or not as severe as studies with other research designs suggested. However, this needs to be investigated in more clinical settings by including sexual orientation as part of the routine assessment. Moreover, adequate sexual-minority specific competencies are important in any case, not just to prevent that sexual minority patients benefit less from treatment.

## Introduction

Compared to heterosexual individuals, lesbian, gay, bisexual (LGB), and other sexual minority individuals are at increased risk for mental disorders and suicide [1], most likely explained by stigma-related stressors that are specific for sexual minorities [2]. The increased risk for mental health problems make LGB individuals a target group for mental health interventions. However, a wealth of literature suggests that sexual minority patients may experience different

**Data Availability Statement:** Data cannot be shared publicly without restrictions because it includes detailed patient level data and anonymity of patients is not guaranteed. However, data for the

unadjusted main analysis without potentially identifying information (age, diagnosis, length of stay etc.) are available via the Open Science foundation (OSF) https://osf.io/py2wn/. Author RM had privileged access in accessing the raw data, which is property of the VAMED Klinik Bad Grönenbach, Sebastian-Kneipp-Allee 3-5 87730 Bad Grönenbach, Germany. However, other researchers may access the raw data by utilizing the following contact information for requests: https://www.vamed.com/de/kontakt/kontakt-projektanfrage/.

**Funding:** The authors received no specific funding for this work.

**Competing interests:** The authors have declared that no competing interests exist.

forms of stigma in health-care services. On an interpersonal level, sexual minority patients report enacted stigma by health care providers, including implicit biases, microaggression, silencing sexual orientation issues, harsh language/behavior, discrimination, rejection, denial of service, and even attempts to change sexual orientation [3–15]. On a structural level, there is often no training in sexual minority-specific competencies in curricula of health care providers, resulting in difficulties providing sexual minority affirmative health care [4, 10, 16–19].

Due to this enacted and structural stigma in healthcare, sexual minority patients may be delivered suboptimal care, resulting in poorer treatment outcome, compared to heterosexual patients. However, most of the studies cited above used selected samples of sexual minority individuals who reported their health care experiences retrospectively. Furthermore, it remains unresolved if enacted and structural stigma actually impairs treatment outcome of sexual minority patients, because most studies did not assess treatment outcome.

The few studies investigating sexual orientation differences in treatment outcome using representative samples or systematically in mental health settings reported mixed results. For example, lesbian and gay patients in the UK were 1.3-times more likely and bisexual patients two times as likely to report long lasting negative effect of psychotherapy, compared to heterosexual patients [20]. In psychological primary care in the UK, lesbian/bisexual women and bisexual men (but not gay men) had poorer outcomes compared to heterosexual patients [21, 22]. In contrast, retrospectively assessed confidence in healthcare and healthcare providers was comparable for heterosexual and gay/bisexual Dutch general practice patients [23]. No sexual orientation differences were found for therapeutic alliance and treatment satisfaction in a study about a substance-abuse rehabilitation program [24]. Finally, an Austrian study found comparable treatment outcome and therapeutic alliance for LGB and heterosexual psychiatric inpatients at risk for suicide [25].

To sum up, there is a wealth of literature suggesting enacted and structural stigma in (mental) healthcare, which likely negatively impacts the quality of treatment and treatment outcome for sexual minority patients. However, the few studies on sexual orientation differences in treatment outcome produced mixed results, perhaps due to methodological differences. Therefore, the main goal of our study is to investigate sexual orientation disparities in treatment outcome and satisfaction with treatment in a mental health care setting not specifically tailored for sexual minority patients, in our case a German inpatient psychiatric clinic.

## Materials and methods

### Participants and procedure

The study sample included patients treated in a German psychiatric "psychosomatic" clinic. In Germany, so-called psychosomatic clinics offer inpatient treatment mostly for patients with different subacute psychiatric disorders. Around 87% of patients are treated in a regular rehabilitation mode (with privately insured patients having more treatment options), and around 13% are treated in an acute mode with more intensive treatment. Most referrals (90%) are made by statutory insurance companies, and around 10% of patients from private insurance companies with referrals made by general practitioners or psychiatrists. Patients treated in this clinic are comparable to other psychosomatic clinics in the region. Important for this research project is that there is no plausible reason to assume that patients are selected by sexual orientation for this clinic.

Indications are mainly psychiatric disorders from the ICD categories F3 (affective disorders), F4 (neurotic, stress-related and somatoform disorders), and F6 (personality disorders). Contraindications mainly include psycho-organic or psychotic disorders, acute substance

dependency, or acute suicidality. The severity of symptoms is medium or severe for the majority of patients, according to the ICD Symptom Rating Scale. The core element of the treatment program is group therapy (mostly two times a week for 90 minutes), accompanied by individual sessions with a psychotherapist (one weekly 30-minute session in rehabilitation and two weekly 50 minutes sessions in acute care). Additional treatment elements include relaxation therapy, physiotherapy, body therapy, disorder specific psychotherapies for anxiety or depression, or occupational therapy in rehabilitation. Psychotherapists are mainly oriented in cognitive behavioral or psychodynamic approaches. Psychopharmacological medication was initiated, continued or modified depending on the psychiatric disorders.

All patients complete an electronic assessment battery within two days after beginning of the treatment and 6–7 days before discharge. Additional observer-based data, such as psychiatric ICD-10 diagnosis, was assessed by the responsible psychotherapist. The sample for this study included 6748 patients admitted to psychiatric departments from December 23, 2013 until November 25, 2020. Of these, 6475 (95%) completed the initial assessment. Both the initial and discharge assessment including the relevant variables were completed by 6032 patients (89% of all and 93% of those who completed the initial assessment). After exclusion of participants who chose the "no declaration" or "other" category in the item on sexual orientation (see below for details), the analytical sample comprised 5609 participants.

All patients gave written consent after an explanation that the results of the assessments were used for routine quality evaluation of treatment and, anonymously, for research purposes. Thus, the data used for this paper is a secondary analysis of data collected for routine quality assessment and was approved by the ethics commission of the University Rostock (Nr. AZ A 2020–0025).

## Measures

**Sexual orientation.** One item about sexual orientation was used in the initial assessment "What describes your sexual orientation best?" with the following 9 response options: 1. heterosexual ("sexually interested in the other sex"), 2. mostly heterosexual, 3. bisexual ("sexually interested in both men and women"), 4. mostly homosexual, 5. homosexual/gay/lesbian (sexually interested in the same sex), 6. asexual (no interest in sexual interactions), 7. I am not sure, 8. other label, 9. no declaration.

**Diagnosis.** The responsible psychotherapist assessed psychiatric diagnosis at admission using the ICD-10 criteria for research. Borderline Personality Disorder (BPD) diagnosis was rated according to DSM-IV criteria to allow comparisons with other studies on BPD. Only the main categories (F1 to F9) were used for the analysis.

**Treatment outcome.** Treatment outcome was defined as changes in levels of well-being and depression, which were assessed at admission and discharge. Well-being was assessed with the related subscale of the HEALTH-49 instrument [26]. We selected this subscale because it has the highest sensitivity for change and is highly correlated with general, transdiagnostic psychological aspects of health-related quality of life, as assessed with the psychological subscale of the SF-8 [26]. Example items are "I feel relaxed", "I feel good", "I can enjoy". The scale ranges from 0 to 20, with lower values meaning more well-being. The reliability was $r_\alpha$ = .87 at admission and $r_\alpha$ = .92 at discharge in our sample.

Depression was assessed with the depression PHQ-9 subscale of the German version of the Patient Health Questionnaire (PHQ-D) [27], which originates from the validated PRIME-MD Patient Health Questionnaire (PHQ) [28]. It has good psychometric properties and is sensitive to change [29]. The scale ranges from 0 to 27, with higher values meaning more depression. The reliability was $r_\alpha$ = .84 at admission and $r_\alpha$ = .88 at discharge in our sample.

**Satisfaction with treatment.** At discharge, satisfaction with treatment was assessed with the psychometrically validated patient satisfaction scale [30]. Example items are "Did you receive the kind of treatment you wanted?" or "Would you recommend our clinic to a friend if he/she needs a similar kind of help?". The scale ranges from 8 to 32, with higher values meaning more satisfaction.

## Data analysis

To quantify treatment outcome, we used pre-post differences (admission vs. discharge) of depression and well-being to create a continuous measure of change, which is the recommended method for natural groups [31]. Sexual orientation differences of treatment outcome and satisfaction with treatment were analyzed with linear regression analyses. For the unadjusted analyses, regression models included sexual orientation and, for change of depression and well-being also baseline levels of depression/well-being to account for regression to the mean and baseline sexual orientation differences. For adjusted analyses, confounding variables were additional predictors in the regression models, including age, length of stay, and diagnosis. We controlled for these potential confounders because they varied by sexual orientation and, for diagnosis, to obtain a trans-diagnostic estimation sexual-orientation differences. We scaled the outcome variables so that the regression coefficients correspond to Cohen's $d$ effect sizes. We interpreted these effect sizes using the usual thresholds [32], i.e., $d < 0.5$ small, $0.5 \leq d < 0.8$ medium, and $d \geq 0.8$ large. All statistical tests were two-sided, with $p < .05$ as significance level. We used R 3.6.3 [33] for statistical analysis, the R-Code is available online via the open science foundation https://osf.io/py2wn/.

## Results

### Sample description

**Sexual orientation.** From a total of 6,038 patients who completed both assessments, 82.3% (88.2) identified as heterosexual, 5.0% (5.3) mostly heterosexual, 1.5% (1.7) bisexual, 0.5% (0.5) mostly gay/lesbian/homosexual, 2.0% (2.2) gay/lesbian/homosexual, 1.0% (1.0) unsure, 0.4% (0.4) other, 0.7% (0.7) asexual, and 6.6% chose the "do not declare" response option. The numbers in brackets refer to the percentages after excluding the "do not declare" responders. Those who did not declare their sexual orientation ($n = 398$) and the few individuals who chose the "other" response option ($n = 25$) were removed from the analysis, leaving 5609 participants for the analytical sample.

**Sexual orientation differences of confounders and baseline levels of depression and well-being.** With respect to age, compared to heterosexual women, bisexual women and women who were unsure about their sexual orientation were statistically significantly and substantially younger (medium or large effects), and mostly heterosexual or lesbian women were statistically significantly younger (small effect sizes) (Table 1). Sexual minority men did not differ statistically significantly from heterosexual men in age.

For length of stay, compared to heterosexual women, bisexual, lesbian, and unsure identified women had significantly longer stays with medium sized effects; mostly heterosexual and asexual identified women had statistically longer stays (small effects). No statistically significant differences in length of stay between sexual minority and heterosexual men were observed.

With respect to diagnoses, some diagnostic groups were very infrequent and not considered for further analyses (F0: 0.2%; F2: 0.2%, F7: 0%, F8: 0.2%, F9: 1.1%). Most notable differences between sexual minority and heterosexual women were observed for F1 (substance use disorders) and F6 (personality disorders), where the proportion of sexual minority women with

**Table 1. Sample description, M (SD) or % (n).**

| | Women | | | | | | | |
|---|---|---|---|---|---|---|---|---|
| | Heterosexual | *Sexual minorities, combined* | Mostly heterosexual | Bisexual | Mostly lesbian | Lesbian | Asexual | Unsure |
| | $n = 3111$ | $n = 454$ | $n = 210$ | $n = 74$ | $n = 24$ | $n = 63$ | $n = 41$ | $n = 42$ |
| Age | 47.98 (10.82) | 43.47 (12.68)** | 45.43 (12.43)** | 36.08 (12.10)** L | 46.54 (10.49) | 43.54 (10.71)** | 49.90 (10.28) | 38.52 (14.12)** M |
| Length of stay | 44.40 (15.94) | 51.66 (20.32)** | 47.34 (18.03)* | 57.47 (21.44)** M | 48.38 (19.86) | 57.21 (21.56)** M | 51.46 (22.10)** | 56.74 (21.24)** M |
| F1 | 15 (462) | 23 (104)** M | 21 (45)* | 28 (21)** L | 17 (4) | 27 (17)** M | 29 (12)* L | 12 (5) |
| F3 | 88 (2741) | 89 (4040) | 89 (187) | 89 (66) | 83 (20) | 83 (52) | 100 (41) | 90 (38) |
| F4 | 44 (1372) | 56 (254)** | 50 (104) | 68 (50)** L | 46 (11) | 52 (33) | 63 (26)* M | 71 (30)** L |
| F5 | 20 (631) | 25 (115)* | 21 (44) | 36 (27)** L | 12 (3) | 27 (17) | 29 (12) | 29 (12) |
| F6 | 13 (419) | 33 (148)** L | 23 (49)** M | 61 (45)** L | 17 (4) | 37 (23)** L | 17 (7) | 48 (20)** L |
| BPD | 6 (183) | 20 (91)** L | 12 (26)** L | 41 (30)** L | 8 (2) | 24 (15)** L | 12 (5) | 31 (13)** L |
| Well-being admission | 2.85 (0.70) | 2.92 (0.66) | 2.85 (0.68) | 2.95 (0.63) | 2.88 (0.61) | 2.80 (0.66) | 3.23 (0.61)** M | 3.09 (0.61)* |
| Well-being discharge | 1.91 (0.84) | 2.09 (0.83)** | 2.03 (0.87)* | 2.11 (0.77)* | 2.01 (0.63) | 1.95 (0.77) | 2.44 (0.93)** | 2.28 (0.77)** |
| Depression admission | 14.72 (5.49) | 15.64 (5.56)** | 15.08 (5.52) | 16.51 (5.32)** | 14.33 (5.65) | 13.43 (5.33) | 18.46 (5.12)** | 18.24 (4.79)** |
| Depression discharge | 9.03 (5.46) | 10.43 (5.78)** | 9.99 (5.57)* | 11.00 (5.98)** | 9.21 (4.82) | 8.60 (4.63) | 12.93 (6.75)** | 12.62 (6.24)** |
| | Men | | | | | | | |
| | Heterosexual | *All sexual minorities, combined* | Mostly heterosexual | Bisexual | Mostly gay | Gay | Asexual | Unsure |
| | $n = 1856$ | $n = 188$ | $n = 89$ | $n = 19$ | $n = 4$ | $n = 59$ | $n = 1$ | $n = 16$ |
| Age | 49.12 (10.38) | 47.39 (10.67)* | 48.78 (10.74) | 48.32 (12.28) | 47.75 (11.73) | 45.00 (8.71)** | 57.00 (-) | 46.75 (14.14) |
| Length of stay | 43.85 (16.05) | 44.70 (16.94) | 43.74 (16.07) | 49.74 (22.2) | 51.00 (22.91) | 46.24 (16.58) | 35.0 (-) | 37.38 (13.51) |
| F1 | 20 (364) | 23 (43) | 27 (24) | 32 (6) | 0 (0) | 17 (10) | 0 (-) | 19 (3) |
| F3 | 87 (1682) | 89 (168) | 89 (79) | 95 (18) | 75 (3 | 90 (53) | 100 (-) | 88 (14) |
| F4 | 39 (718) | 43 (80) | 37 (33) | 42 (8) | 75 (3) | 49 (29) | 100 (-) | 38 (6) |
| F5 | 17 (324) | 18 (33) | 20 (18) | 16 (3) | 0 (0) | 15 (9) | 0 (-) | 19 (3) |
| F6 | 12 (218) | 20 (37)** M | 18 (16) | 32 (6)* L | 25 (1) | 20 (12)* M | 0 (-) | 12 (2) |
| BPD | 2 (45) | 7 (14)** L | 6 (5)* L | 11 (2) | 0 (0) | 8 (5)** L | 0 (-) | 12 (2)* L |
| Well-being admission | 2.77 (0.72) | 2.88 (0.71) | 2.94 (0.62)* | 2.81 (0.66) | 3.15 (0.25) | 2.82 (0.86) | 3.20 (-) | 2.81 (0.81) |
| Well-being discharge | 1.80 (0.86) | 2.01 (0.94)** | 1.99 (0.81)* | 1.76 (1.04) | 2.35 (0.66) | 2.00 (1.11) | 3.60 (-) | 2.28 (0.89)* M |
| Depression admission | 13.76 (5.73) | 15.23 (5.77)** | 15.67 (5.40)** | 15.21 (6.53) | 19.75 (2.50)* | 14.46 (6.13) | 24 (-) | 13.94 (5.41) |
| Depression discharge | 8.18 (5.56) | 9.75 (6.19)** | 9.85 (6.14)** | 9.21 (6.25) | 11.00 (4.32) | 9.48 (6.56) | 23 (-) | 9.69 (5.08) |

M medium effect size,

L large effect size.

For statistically significant findings, effect size estimates are small if not denoted otherwise. No effect size was given in case of insufficient cell-frequencies (e.g., 100% or 0%, or $n = 1$), or if the comparison was not statistically significant.

$p < .05$,

** $p < .01$.

these diagnosis was statistically significantly higher, with medium to large effect sizes, compared to heterosexual women. Only few statistically significant differences in diagnoses occurred between sexual minority and heterosexual men: bisexual and gay identified men had statistically significantly higher proportions diagnosed with personality disorders.

For baseline-levels of depression and well-being, among women, there was only one statistically significant difference with medium effect size: women identified as asexual had lower levels of well-being, compared to heterosexual women. Women unsure of their sexual orientation had statistically significantly higher levels of depression (small effect). Baseline-levels of depression were significantly higher (small effects) for women identified as bisexual, asexual, or unsure of their sexual orientation. Compared to heterosexual men, mostly heterosexual men had statistically significantly lower baseline-levels of well-being and higher levels of base-line depression (both small effects), and mostly gay men had statistically significantly higher levels of base-line depression (large effect).

## Change of depression and well-being, and satisfaction with treatment

**Change of well-being.**   Among women, in unadjusted analysis, all sexual minority sub-groups improved slightly less in well-being, compared to the heterosexual reference group, but all differences were small and only significant for those identified as mostly heterosexual, asexual, or all sexual minority women combined (Tables 2 and 3). The sexual orientation differences further decreased in the adjusted analysis, remaining only statistically significant for women identifying as asexual.

Among men, in unadjusted analysis, those identified as mostly heterosexual, gay, or unsure of their sexual orientation improved slightly less in well-being, compared to the heterosexual reference group, and bisexual men had slightly better improvement. These differences were small, except for men with unsure sexual orientation, where the difference was medium-sized and statistically significant (Tables 2 and 3). Comparable results were found in the adjusted analyses. There were too few men identifying as mostly gay or asexual to allow meaningful analyses.

**Change of depression.**   Among women, in unadjusted analysis, all sexual minority sub-groups improved slightly less in depression, compared to the heterosexual reference group, but all differences were small and only significant for those identifying as mostly heterosexual, asexual, or unsure of their sexual orientation, and all sexual minority women combined (Tables 2 and 3). The sexual orientation differences further decreased in the adjusted analysis, remaining only statistically significant for women identifying as asexual.

Among men, in unadjusted analysis, all sexual minority subgroups improved slightly less in depression, compared to the heterosexual reference group, but these differences were small and only statistically significant for all sexual minority men combined. The adjusted analysis produced similar results and none of the differences was statistically significant.

**Satisfaction with treatment.**   Among women, in unadjusted analyses, sexual minority subgroups were slightly less satisfied with treatment (statistically significant only for mostly heterosexual women), and mostly lesbian and lesbian women were slightly more satisfied (but not statistically significant), compared to heterosexual women (Tables 2 and 4). All differences were small. The adjusted analysis had comparable results, except that only the difference for women identifying as asexual was statistically significant.

Among men, all sexual minority subgroups had lower levels of satisfaction with treatment, and the difference was statistically significant for all sexual minority men combined and for bisexual men. For the latter group, the difference was medium-sized, whereas all other differences were small. The adjusted analysis produced comparable results. (Tables 2 and 4).

**Table 2. Treatment outcome and satisfaction with treatment by sexual orientation, *M (SD)*.**

| | Women | | | | | | | |
|---|---|---|---|---|---|---|---|---|
| | Heterosexual | *All sexual minorities combined* | Mostly heterosexual | Bisexual | Mostly lesbian | Lesbian | Asexual | Unsure |
| | N = 3111 | n = 362 | n = 210 | n = 74 | n = 24 | n = 63 | n = 41 | n = 42 |
| Change in well-being (difference admission—discharge | 0.94 (0.82) | 0.82 (0.83) | 0.81 (0.81) | 0.84 (0.77) | 0.88 (0.67) | 0.85 (0.93) | 0.79 (0.94) | 0.81 (0.87) |
| Change in depression (difference admssion—discharge) | 5.54 (5.15) | 5.21 (5.19) | 5.09 (4.92) | 5.51 (5.14) | 5.12 (4.79) | 4.83 (5.32) | 5.54 (6.08) | 5.62 (5.86) |
| Treatment satisfaction | 25.56 (4.80) | 25.10 (4.87) | 24.81 (4.82) | 24.77 (4.99) | 26.62 (4.20) | 26.67 (4.31) | 24.17 (6.14) | 24.79 (4.20) |
| | Men | | | | | | | |
| | Heterosexual | *All sexual minorities combined* | Mostly heterosexual | Bisexual | Mostly gay | Gay | Asexual | Unsure |
| | n = 1856 | n = 188 | n = 89 | n = 19 | n = 4 | n = 59 | n = 1 | n = 16 |
| Change in well-being (difference admission—discharge | 0.98 (0.81) | 0.88 (0.85) | 0.95 (0.77) | 1.05 (1.05) | 0.80 (0.78) | 0.82 (0.92) | 0.40 (-) | 0.54 (0.67) |
| Change in depression (difference admssion—discharge) | 5.58 (4.98) | 5.48 (5.07) | 5.82 (4.84) | 6.00 (5.60) | 8.75 (6.60) | 4.98 (5.33) | 1.00 (-) | 4.25 (4.34) |
| Treatment satisfaction | 25.98 (4.64) | 25.06 (5.05) | 25.40 (4.82) | 23.42 (6.23) | 29.00 (2.58) | 25.00 (5.35) | 16.00 (-) | 24.94 (3.11) |

**Table 3. Treatment outcome by sexual orientation—Results of unadjusted and adjusted regression analyses, *β (SE)*.**

| | Women | | | | Men | | | |
|---|---|---|---|---|---|---|---|---|
| | Well-being | | Depression | | Well-being | | Depression | |
| | Unadjusted | Adjusted | Unadjusted | Adjusted | Unadjusted | Adjusted | Unadjusted | Adjusted |
| Sexual Orientation | | | | | | | | |
| *All sexual minorities combined* | -0.18 (0.05)** | -0.08 (0.05) | -0.16 (0.04)** | -0.06 (0.04) | -0.16 (0.07)* | -0.11 (0.07) | -0.13 (0.07)* | -0.10 (0.07) |
| Mostly heterosexual | -0.15 (0.07)* | -0.09 (0.06) | -0.13 (0.06)* | -0.08 (0.06) | -0.11 (0.10) | -0.08 (0.10) | -0.10 (0.09) | -0.08 (0.09) |
| Bisexual | -0.18 (0.11) | -0.08 (0.11) | -0.17 (0.11) | 0.07 (0.11) | 0.08 (0.21) | 0.16 (0.21) | -0.03 (0.20) | 0.02 (0.20) |
| Mostly lesbian/gay | -0.10 (0.19) | -0.12 (0.18) | -0.06 (0.18) | -0.07 (0.18) | - | - | - | - |
| Lesbian/gay | -0.08 (0.12) | -0.01 (0.11) | -0.04 (0.11) | 0.05 (0.11) | -0.21 (0.12) | -0.13 (0.12) | -0.17 (0.12) | -0.12 (0.11) |
| Asexual | -0.41 (0.14)** | -0.35 (0.14)* | -0.34 (0.14)* | -0.30 (0.14)* | - | - | - | - |
| Unsure | -0.30 (0.27) | -0.10 (0.14) | -0.30 (0.14)* | -0.12 (0.14) | -0.55 (0.23)* | -0.50 (0.23)* | -0.28 (0.22) | -0.22 (0.30) |
| Well-being/depression at admission | 0.57 (0.02)** | 0.63 (0.02)** | 0.09 (0.00)** | 0.10 (0.00)** | 0.51 (0.03)** | 0.55 (0.03)** | 0.08 (0.00)** | 0.08 (0.00)** |
| Age | | 0.00 (0.00)* | | 0.00 (0.00)* | | 0.01 (0.00)** | | 0.00 (0.00)* |
| Length of stay | | 0.00 (0.00)** | | 0.00 (0.00)* | | 0.01 (0.00)** | | 0.01 (0.00)** |
| F1 | | 0.04 (0.04) | | 0.03 (0.04) | | 0.01 (0.05) | | -0.02 (0.05) |
| F3 | | -0.41 (0.05)** | | -0.28 (0.05)** | | -0.36 (0.07)** | | -0.30 (0.07)** |
| F4 | | -0.22 (0.03) ** | | -0.22 (0.03)** | | -0.24 (0.05)** | | -0.25 (0.04)** |
| F5 | | -0.05 (0.04) | | -0.05 (0.04) | | -0.15 (0.05)** | | -0.20 (0.05)** |
| F6 | | -0.48 (0.05)** | | -0.42 (0.05)** | | -0.43 (0.07)** | | -0.30 (0.06)** |

Note: The quantitative measures were scaled to allow effect-size interpretation (Cohens's *d*). Missing entries indicate that the frequency in the cell was too low for multivariate regression analysis. Results for *n* = 1 subgroups not given. Adjusted analysis included age, length of stay, and diagnosis as predictors. Results for confounders are for the regression model with full information on sexual orientation.

* $p < .05$,

** $p < .01$.

**Table 4. Satisfaction with treatment by sexual orientation—Results of unadjusted and adjusted regression analyses, *β (SE)*.**

| | Women | | Men | |
|---|---|---|---|---|
| | **Unadjusted** | **Adjusted** | **Unadjusted** | **Adjusted** |
| Sexual Orientation | | | | |
| *All sexual minorities combined* | -0.10 (0.05) | -0.06 (0.05) | -0.20 (0.08)** | -0.15 (0.07)* |
| Mostly heterosexual | -0.16 (0.07)* | -0.12 (0.07) | -0.12 (0.11) | -0.08 (0.10) |
| Bisexual | -0.17 (0.12) | -0.01 (0.12) | -0.54 (0.23)* | -0.51 (0.22)* |
| Mostly lesbian/gay | 0.22 (0.21) | 0.18 (0.20) | - | - |
| Lesbian/gay | 0.23 (0.13) | 0.19 (0.12) | -0.21 (0.13) | -0.15 (0.13) |
| Asexual | -0.29 (0.16) | -0.34 (0.15)* | - | - |
| Unsure | -0.16 (0.16) | -0.10 (0.15) | -0.22 (0.25) | -0.10 (0.24) |
| Age | | 0.00 (0.00)* | | 0.01 (0.00)** |
| Length of stay | | 0.02 (0.00)** | | 0.02 (0.00)** |
| F1 | | -0.10 (0.05)* | | -0.10 (0.05) |
| F3 | | -0.20 (0.05)** | | -0.20 (0.07)** |
| F4 | | -0.09 (0.04)* | | -0.23 (0.05)** |
| F5 | | 0.02 (0.04) | | -0.11 (0.06)* |
| F6 | | -0.63 (0.06)** | | -0.37 (0.07)** |

Note: The quantitative measures were scaled to allow effect-size interpretation. Missing entries indicate that the frequency in the cell was zero or too low for regression analysis. Adjusted analysis included age, length of stay, and diagnosis as predictors. Results for confounders are for the regression model with full information on sexual orientation.

\* $p < .05$,

\*\* $p < .01$.

## Discussion

The goal of our study was to compare if sexual minority patients have less optimal treatment outcome and satisfaction with treatment than heterosexual patients in a psychotherapeutic inpatient setting, as would be expected given the stigma processes described in the literature. Contrary to our hypothesis, most of the sexual orientation differences in treatment outcome (improvement of depression and well-being) and satisfaction with treatment were very small (d < 0.20) and lacked statistical significance. However, for most comparisons, the sign of the difference indicated a slightly less optimal treatment outcome and a slightly lower satisfaction with treatment for sexual minority patients. Furthermore, two differences achieved a medium effect size (d > 0.50): less improvement of well-being for men unsure of their sexual orientation and lower treatment satisfaction for bisexual men.

Since most of the sexual orientation differences were small and lacked statistical significances, it is not possible to provide a binary decision if sexual minority patients benefit similar or less from treatment than heterosexual patients. On the one hand, it could be argued that the differences are too small to deem them problematic. On the other hand, even small differences should not be ignored, and uncertainty remains for the smaller subgroups of sexual minority patients. Only very large samples can provide exact estimations of smaller differences. Nevertheless, our findings suggest that the differences are not universal or perhaps not as large as suggested in the literature. This is in line with recent findings from Austria and, for men, from the UK [21, 25]. More related research is clearly needed, and we recommend assessing sexual orientation in the quality assessments in all treatment settings. We experienced that this is possible and feasible in our study, in line with studies finding that nearly all patients are willing to report their sexual orientation and believe that such questions are important [25, 34–38].

Furthermore, future studies should also assess sexual minority specific competencies of healthcare providers and its impact on treatment outcome of sexual minority patient. Several guidelines and recommendations discuss how to achieve adequate sexual-minority-specific competency in health-care settings [39–42]. Trainings and workshops in the curriculum and at the workplace are common suggestions [42], but see Dean et al. [10] for a critical account. LGB affirmative leadership and policies may be important [43] as well as staff members who come out as sexual minority and who initiate awareness and changes in attitudes and knowledge of other staff members [43–45]. We also would like to stress that adequate sexual-minority specific competencies are important in any case, not just to prevent that sexual minority patients benefit less from treatment. Finally, it is important to point out that, given the long history of pathologization of homosexuality in German psychiatry [46], our findings are far from being self-evident.

## Limitations

Despite the large sample size, some sexual minority subgroups were too small to draw firm conclusions, with the risk of false-negatives and false-positive sexual orientation differences. This is a general challenge in sexual minority research that goes beyond the gay/straight binary. The problem is even greater when considering intersectionality, that is, individuals with more than one minority attributes (for example lesbian women who migrated to Germany and who have a physical disability). Our findings may thus not apply to members of these varying intersections. However, it needs extremely large samples to allow analysis of intersecting minorities. A related problem are confounding variables. We found that length of stay was associated with better treatment outcome and that personality disorder diagnoses were associated with worse treatment outcome. These confounders sometimes differed by sexual orientation, and adjusting in multivariate analyses reduced sexual orientation disparities. However, we may have missed other potentially important confounding variables.

Different forms of biases are a general challenge in sexual minority research and may have impacted our results, too. For example, information bias occurs when sexual minority individuals do not disclose in surveys [47]. This could lead to an underestimation of sexual orientation differences in treatment outcome if those who falsely identify as heterosexual benefit less from treatment than actual heterosexual patients. Another problem could be selection bias, for example if heterosexual and sexual minority individuals who decide to enter treatment differ in variables which impact treatment outcome and if these variables are not assessed and controlled in multivariate analysis. Furthermore, our study and other related research may underestimate sexual orientation differences in treatment outcome across providers if the decision of a provider to include a measure of sexual orientation in the assessment correlates with sexual minority competency of the provider. Therefore, many more studies in different mental health care settings are necessary to draw firm conclusions.

A reviewer of a previous version of this paper wondered how the different treatment elements or severity of disorders confound the results. We do not think this is a problem because there is no reason to assume that treatment varied by sexual orientation, and baseline symptoms were controlled for in the statistical analyses. Diagnostic biases could have decreased the validity of our results. Experimental studies using case-vignettes found that clinicians more likely diagnose patients with BPD when the vignette included information about a gay or lesbian sexual orientation [48]. Indeed, in our sample, sexual minority patients were more likely diagnosed with personality disorders. Alternatively, this could be a valid finding because sexual minority individuals experience above-average stressors early in life [49–51], perhaps leading to an increased risk for developing personality disorders.

Our choice of sexual orientation labels may not be sufficient for some sexual minority individuals who may have preferred other categories, and a single item on sexual orientation seems difficult to understand for some people [52]. Furthermore, we did not assess different dimensions of sexual orientation in detail (attraction, identity, behavior). This can be problematic since a substantial fraction of individuals with same-sex sexual behavior identifies as heterosexual [53, 54]. We also did not assess gender minority and intersex status. However, our study is one of the few that also included other sexual orientation labels than LGB.

## Conclusions

Contrary to our expectations, we found no or mostly small sexual orientation differences for most sexual minority subgroups in treatment outcome and satisfaction with treatment in a German psychiatric inpatient setting. These findings need to be replicated in other health-care settings to draw firm conclusions about the actual problem of barriers in mental health care for sexual minority patients. Furthermore, adequate sexual-minority specific competencies are important in any case, not just to prevent that sexual minority patients benefit less from treatment.

## Author Contributions

**Conceptualization:** Martin Plöderl, Robert Mestel, Clemens Fartacek.

**Data curation:** Martin Plöderl, Robert Mestel.

**Formal analysis:** Martin Plöderl.

**Methodology:** Robert Mestel.

**Supervision:** Robert Mestel.

**Validation:** Robert Mestel.

**Writing – original draft:** Martin Plöderl, Robert Mestel, Clemens Fartacek.

**Writing – review & editing:** Martin Plöderl, Robert Mestel, Clemens Fartacek.

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
