## [Decision Letter · Decision Letter 0]

12 Oct 2021

PONE-D-21-24975Differences by Sexual Orientation in Treatment Outcome and Satisfaction with Treatment among Inpatients of a German Psychosomatics ClinicPLOS ONE

Thank you for submitting your manuscript to PLOS ONE. After careful consideration, we feel that it has merit but does not fully meet PLOS ONE’s publication criteria as it currently stands. Therefore, we invite you to submit a revised version of the manuscript that addresses the points raised during the review process.

We look forward to receiving your revised manuscript.

Kind regards,

Luigi Lavorgna

Academic Editor

PLOS ONE

Journal Requirements:

2.Please indicate whether all patient data was anonymized prior to your access and analysis

Reviewers' comments:

Reviewer's Responses to Questions

**Comments to the Author**

1. Is the manuscript technically sound, and do the data support the conclusions?

Reviewer #1: Yes

Reviewer #2: Yes

2. Has the statistical analysis been performed appropriately and rigorously? 

Reviewer #1: No

Reviewer #2: Yes

3. Have the authors made all data underlying the findings in their manuscript fully available?

Reviewer #1: No

Reviewer #2: Yes

4. Is the manuscript presented in an intelligible fashion and written in standard English?

Reviewer #1: Yes

Reviewer #2: Yes

5. Review Comments to the Author

Reviewer #1: Plöderl and colleagues reported on associations between sexual orientation and treatment outcome and satisfaction among inpatients of a German psychosomatics clinic. The manuscript is overall clear and well written. Methods are sound. The topic is interesting and definitely worth being investigated. However, I have some issues I would like the authors comment on.

“Psychosomatic” is a rather general term and not widely accepted anymore. Are authors referring to functional disorders?

Why are authors referring to LGB and not to LGBT? The evidence reported in the first paragraph of the introduction also applies to the T component of the LGBT community.

In the introduction, among previous studies, authors should also refer to Lavorgna et al. Mult Scler Relat Disord. 2017.

Authors state they evaluated inpatients. I just want to double check they are referring to patients admitted to hospital? It is rather atypical admitting psychosomatic patients to hospital (they are generally seen in outpatients and/or day care services).

Have authors tried to compare heterosexual vs other groups? I believe all other groups (mostly heterosexual, bisexual, gay, mostly gay, mostly lesbian, lesbian, asexual, unsure) could be combined and directly compared to heterosexuals. This should increase the statistical power, while sub-analyses for different groups could be commented at descriptive level.

Reviewer #2: Intersting work methodologically correct despite the very small sample size of no hetero sexually oriented patients resulting from the interview. Limitations of the study are well illustrated, could the authors refer to some historical case sseries with similar charactheristics to reinforce their observations, apart from the studies listed in the introduction?

Lenght of stay from admission to discharge is an important discriminant, please try to give it more prominence in the discussion.

Table 3 is heavy to read, I suggest to ameliorate it.

6. PLOS authors have the option to publish the peer review history of their article (what does this mean?). If published, this will include your full peer review and any attached files.

Reviewer #1: No

Reviewer #2: No

---

## [Author Response · Author response to Decision Letter 0]

6 Dec 2021

Dear Reviewers!

Thank you for reviewing our manuscript and your thoughtful comments. We pasted your comments and respond to each pointwise. We hope we have addressed your suggestions accordingly. 

Reviewer #1: 

Plöderl and colleagues reported on associations between sexual orientation and treatment outcome and satisfaction among inpatients of a German psychosomatics clinic. The manuscript is overall clear and well written. Methods are sound. The topic is interesting and definitely worth being investigated.

Reply: Thank you for the positive evaluation. 

However, I have some issues I would like the authors comment on.

“Psychosomatic” is a rather general term and not widely accepted anymore. Are authors referring to functional disorders?

Reply: We agree, this needs some more clarification for readers outside of the German region. We now included some explanation in the participants/procedure section: “The study sample included patients treated in a German psychiatric ‘psychosomatic’ clinic. In Germany, so-called psychosomatic clinics offer inpatient treatment mostly for patients with different subacute psychiatric disorders.”

Why are authors referring to LGB and not to LGBT? The evidence reported in the first paragraph of the introduction also applies to the T component of the LGBT community.

Reply: We agree that some studies in the introduction also included transgender (trans) individuals in their samples. However, most studies did not include trans individuals or did not assess trans identities. We decided to only refer to LGB because sexual minorities were the focus of our study. Including trans issues would go beyond the scope of our paper. Furthermore, in some areas, healthcare-barriers for sexual minorities differ substantially from those of trans people (e.g., access to gender affirming hormonal treatment or surgeries). This is why trans activists and researchers often criticize the combination of sexual minorities and gender minorities in studies. Unfortunately, we did not assess transgender identities in our sample. This was addressed in the limitation section. 

In the introduction, among previous studies, authors should also refer to Lavorgna et al. Mult Scler Relat Disord. 2017.

Reply: Thank you for making us aware of this important study. Lavorgna et al. reported less frequent use of psychological service among LGB patients (no trans patients in the sample) with multiple sclerosis, and also that switching multiple-sclerosis services was associated with experienced homophobic behavior in these services. Disturbing findings were that 10 out of the 35 participating LGB patients experienced insensitive or hurtful comments about their sexual orientation, and 4 even got the advice to change their sexual orientation. We therefore cited this study in the section where we summarize the evidence about homophobic experiences in healthcare (end of first paragraph). As a note, the sampling procedure in this study may have introduced biases, and the sample of LGB patients was small, thus our criticism of existing studies in the introduction also applies to this study. 

Authors state they evaluated inpatients. I just want to double check they are referring to patients admitted to hospital? It is rather atypical admitting psychosomatic patients to hospital (they are generally seen in outpatients and/or day care services).

Reply: This is correct, all patients were inpatients. As described in the method section, the symptom severity is medium or severe for most patients. 

Have authors tried to compare heterosexual vs other groups? I believe all other groups (mostly heterosexual, bisexual, gay, mostly gay, mostly lesbian, lesbian, asexual, unsure) could be combined and directly compared to heterosexuals. This should increase the statistical power, while sub-analyses for different groups could be commented at descriptive level.

Reply: Yes, the results for sexual minorities combined were already given in the original submission, right after the results for heterosexuals (see the second columns in Tables 1 and 2, and the second lines in Tables 3 and 4). We now italicized these columns/lines to enhance visibility.

We see a problem in presenting the results for individual subgroups only descriptively, that is, without adjustment of confounding variables, because these confounders are often substantially different in these subgroups (see comments below). Besides, many of the subgroups are large enough to allow adjusted group comparisons. 

Reviewer #2

Intersting work methodologically correct despite the very small sample size of no hetero sexually oriented patients resulting from the interview. Limitations of the study are well illustrated,

Reply: Thank you for the positive evaluation. We agree that the number of individuals is low for some sexual minority subgroups. However, compared to many related studies, our sample size is actually quite large. Furthermore, many of the subgroups are large enough to allow adjusted group comparisons. We plan to repeat the analysis in 5 to 10 years to see if the findings are robust for the currently smaller sexual minority groups. 

 could the authors refer to some historical case sseries with similar charactheristics to reinforce their observations, apart from the studies listed in the introduction?

Reply: Thank you for the suggestion to include historical information from German psychiatry. We are not aware of a study about case series with similar characteristics. As Stakelbeck and Frank pointed out: “What is known about the actual experience of gays and lesbians who are seeking psychiatric or psychotherapeutic treatment in Germany? Surprisingly, there is hardly any literature and virtually no empirical studies on the subject.”

Stakelbeck, F., & Frank, U. (2003). From Perversion to Sexual Identity: Concepts of Homosexuality and Its Treatment in Germany. Journal of Gay & Lesbian Psychotherapy, 7(1-2), 23–46. doi:10.1300/j236v07n01_03 

Stakelbeck and Frank gave an excellent overview of the history of homosexuality in psychiatry and psychotherapy in German, and we now cite this paper in the discussion. We now also discuss that, given the historical background, it is not self-evident that we found no or only minor differences between heterosexual and sexual minority patients. 

Lenght of stay from admission to discharge is an important discriminant, please try to give it more prominence in the discussion.

Reply: Length of stay was indeed an independent predictor of treatment outcome and satisfaction with treatment (as were diagnoses, especially personality disorders). There were some sexual orientation differences for length of stay and diagnoses, especially the diagnosis of a personality disorder, thus it was important to adjust for these variables. However, length of stay (and other confounders) did not seem to be differentially associated with treatment outcome for the sexual orientation subgroups. Thus, we think that discussing these variables in more depth goes beyond the scope of the paper. However, we now discuss length of stay and personality disorders as important confounders in the limitation section: “A related problem are confounding variables. We found that length of stay was associated with better treatment outcome and personality disorder diagnoses were associated with worse treatment outcome. These confounders sometimes differed by sexual orientation, and adjusting in multivariate analyses reduced sexual orientation disparities. However, we may have missed other potentially important confounding variables”

Table 3 is heavy to read, I suggest to ameliorate it.

Reply: we sorted out different ways to simplify this table. For example, the results for confounding variables could be moved to an online supplement. However, perhaps many readers would like to see the results for important confounders in contrast to sexual orientation variables (to bring the effect sizes into context). Another option would be to split the table for women and men, or for adjusted and unadjusted analysis. However, these changes of formatting would make comparisons between gender or adjusted/unadjusted analysis more complicated. Thus, we would like to leave the table as it is. Of course we are open to make changes if you think this is really important.

---

## [Editor Report · Decision Letter 1]

10 Jan 2022

Differences by Sexual Orientation in Treatment Outcome and Satisfaction with Treatment among Inpatients of a German Psychiatric Clinic

PONE-D-21-24975R

We’re pleased to inform you that your manuscript has been judged scientifically suitable for publication and will be formally accepted for publication once it meets all outstanding technical requirements.

Kind regards,

Luigi Lavorgna

Academic Editor

PLOS ONE
---

## [Editor Report · Acceptance letter]

12 Jan 2022

PONE-D-21-24975R1 

Differences by sexual orientation in treatment outcome and satisfaction with treatment among inpatients of a German psychiatric clinic 

Dear Dr. Plöderl:

I'm pleased to inform you that your manuscript has been deemed suitable for publication in PLOS ONE. Congratulations! Your manuscript is now with our production department. 

Kind regards, 

on behalf of

Dr. Luigi Lavorgna 

Academic Editor

PLOS ONE